# Evidence of a Positive Link between Consumption and Supplementation of Ascorbic Acid and Bone Mineral Density

**DOI:** 10.3390/nu13031012

**Published:** 2021-03-21

**Authors:** Mariangela Rondanelli, Gabriella Peroni, Federica Fossari, Viviana Vecchio, Milena Anna Faliva, Maurizio Naso, Simone Perna, Enrica Di Paolo, Antonella Riva, Giovanna Petrangolini, Mara Nichetti, Alice Tartara

**Affiliations:** 1IRCCS Mondino Foundation, 27100 Pavia, Italy; mariangela.rondanelli@unipv.it; 2Department of Public Health, Experimental and Forensic Medicine, University of Pavia, 27100 Pavia, Italy; 3Endocrinology and Nutrition Unit, Azienda di Servizi alla Persona “Istituto Santa Margherita”, University of Pavia, 27100 Pavia, Italy; federica.fossari01@universitadipavia.it (F.F.); viviana.vecchio01@universitadipavia.it (V.V.); milena.faliva@gmail.com (M.A.F.); mau.na.mn@gmail.com (M.N.); dietista.mara.nichetti@gmail.com (M.N.); alice.tartara01@universitadipavia.it (A.T.); 4Department of Biology, College of Science, University of Bahrain, 32038 Sakhir, Bahrain; simoneperna@hotmail.it; 5General Geriatric Unit, Azienda di Servizi alla Persona “Istituto Santa Margherita”, 27100 Pavia, Italy; enrica_dipaolo@asppavia.it; 6Research and Development Department, Indena SpA, 20139 Milan, Italy; antonella.riva@indena.com (A.R.); giovanna.petrangolini@indena.com (G.P.)

**Keywords:** ascorbic acid, osteoporosis, supplementation, nutrients, bone health

## Abstract

In animal models it has been shown that ascorbic acid (AA) is an essential cofactor for the hydroxylation of proline in collagen synthesis. However, there are still no precise indications regarding the role of AA in maintaining bone health in humans, so the aim of this narrative review was to consider state of the art on correlation between bone mineral density (BMD), AA dietary intake and AA blood levels, and on the effectiveness of AA supplement in humans. This review included 25 eligible studies. Fifteen studies evaluated correlations between AA intake and BMD: eight studies demonstrated a positive correlation between AA dietary intake and BMD in 9664 menopausal women and one significant interaction between effects of AA intake and hormone therapy. These data were also confirmed starting from adolescence (14,566 subjects). Considering studies on AA blood concentration and BMD, there are four (337 patients) that confirm a positive correlation. Regarding studies on supplementation, there were six (2671 subjects), of which one was carried out with AA supplementation exclusively in 994 postmenopausal women with a daily average dose of 745 mg (average period: 12.4 years). BMD values were found to be approximately 3% higher in women who took supplements.

## 1. Introduction

Ascorbic acid (AA), or vitamin C, is a water-soluble vitamin responsible for the biosynthesis of collagen, neurotransmitters, and L-carnitine. In addition, AA also plays an important action on the immune function system and for iron absorption. The antioxidant effects of AA supplementation showed during the past decades clear effect on cancers, cardiovascular diseases prevention [1].

Fresh fruits and vegetables are the richest sources of AA, in particular peppers, kiwis, citrus fruits, strawberries, tomatoes and green leafy vegetables (such as field chicory and broccoli); this vitamin is then added as an ingredient in many fruit juice-based drinks [2]. The average dietary intake of AA (60–100 mg/day) reflects a plasma level of approximately 40–60 µM. The relationship between the dose taken and the plasma concentration of this vitamin was found to be of the sigmoidal type: intakes higher than 100 mg/day are beyond the linear portion of the curve, with plasma saturation and circulating cells for a dose between 200 and 400 mg/day, corresponding to concentrations of about 70–85 µM, with consequent urinary excretion of the vitamin [3]. AA can be considered deficient for plasma levels below 11 µmol/L, with the appearance of biochemical and/or clinical symptoms [4]. The (RDA) of AA for adult is 90 mg/day in males and 75 mg/day in females; 15–65 mg/day for children depending on age [5]. An, higher amount is advised during the pregnancy, lactation and for smokers [5]. Based on data collected through the National Health and Nutrition Examination Survey (NHANES) 2003–2004, an overall prevalence of AA deficiency in the United States of 7.1 ± 0.9% was calculated. Defined as serum AA levels <11.4 µmol L, the deficiency was significantly more frequent in males (8.2%) than in females (6.0%); however, no difference was found between different races/ethnic groups. The deficiency was observed less commonly in children (6–11 years of age) and older adults (≥60 years of age) than in young adults (20–39 years) [6].

### AA and Bone: In Vitro and Animal Model Studies

Table 1 synthesizes pathophysiological key messages on this topic.

## 2. Materials and Methods

The review followed the guidelines provided by Egger et al. [20]. The review considered as main research question “state of the art on correlation between BMD and AA dietary intake and AA blood levels and on the effectiveness of AA supplement in humans”. An extensive literature search was done in Scopus. PubMed and Google Scholar following the through the revision of articles on correlation between BMD and AA dietary intake and AA blood levels and on the effectiveness of AA supplementation in humans. Figure 1 shows the flow chart of the studies considered for this narrative review.

## 3. Results

### 3.1. Hematic Values of AA and Bone Metabolism

This research was conducted based on the keywords: “blood AA” AND “bone” AND “humans”. For the present part we have analyzed a total of four studies: three case-control studies and one cross-sectional study. Table 2 shows the studies that consider the relationship between blood levels of AA and BMD and fracture risk.

### 3.2. Dietary Intake of AA and Bone Metabolism

This research was conducted based on the keywords: “AA intake” AND “bone” AND “humans”. For the present part we have analyzed a total of 15 studies: five cross-sectional studies, four cohort studies, three case-control studies, one observational study, one longitudinal study and one randomized, double-blind, placebo-controlled study. Table 3 shows the studies that evaluated the relationship between dietary intake of AA and bone metabolism.

### 3.3. Supplementation with AA and Bone Metabolism

This research was conducted based on the keywords: “ascorbic supplementation” AND “bone” AND “humans”. For the present part we have analyzed a total of six studies: two randomized controlled study, one cohort study, one observational study, one randomized, double-blind, controlled trial and one randomized, double-blind, placebo- controlled study. Table 4 shows the studies that evaluated the relations between AA supplementation and bone metabolism.

## 4. Discussion

### 4.1. Plasma Levels of AA and Bone Metabolism

As for plasma levels of AA, Maggio and colleagues, in their cross-sectional case-control study conducted on 150 postmenopausal women aged ≥ 60 years, observed plasma levels of antioxidants, including AA, that were significantly lower in participants suffering from osteoporosis with T-score ≤ −3.5 (cases, *n* = 75) compared to healthy controls (T-score ≥ −1, *n* = 75) [21]. Several other case-control studies then investigated the existence of a relationship between hip fractures or other fragility fractures and serum concentrations of AA. By measuring these concentrations in a group of 20 patients aged ≥ 70 years, admitted to hospital for new fracture of the femoral or intertrochanteric neck, and comparing them with those of as many healthy controls matched for age, sex and time of detection, significantly lower values emerged in the first group compared to the second (34 ± 19 μmoL/L vs. 54 ± 30 μmoL/L). According to the authors, the reduction in serum concentrations of AA could be part of a more general nutritional insufficiency that characterizes elderly patients with hip fractures [22]. Similarly, in a 2007 Spanish study, 167 patients aged ≥ 65 years with osteoporotic fractures were compared with 167 subjects without a fracture diagnosis, matched for age and gender. The results showed a statistically significant difference in blood levels of AA, which was higher in controls; in addition, a linear association was found between serum AA and fracture risk, significantly reduced in subjects with serum values in the upper quartile. On the other hand, no relationship was highlighted between dietary intake of AA (evaluated by semi-quantitative food frequency questionnaire, FFQ) and risk of fractures [23]. On the contrary, a further study conducted in 2001 attempted to investigate the relationship between fracture risk and various blood parameters, including AA, in elderly women with hip fractures, compared to a control group. It was possible to highlight how women with hip fracture had higher serum AA values (42.7 ± 21.4 μmoL/L) than “healthy” controls (20.8 ± 14.2 μmoL/L) [24].

Regarding the dietary intake of the participants, the AA was introduced only by foods in three studies, while the supplements were taken under consideration in the study of Martinez et al. [23] All the patients had no AA deficiency and AA blood levels were all within normal ranges, even if there are differences between the groups.

### 4.2. Intake of AA and Effects on Bone Metabolism in Humans

Several epidemiological studies have investigated the possible association between BMD and dietary intake of AA, particularly in postmenopausal women. A cross-sectional analysis conducted on 775 postmenopausal women, aged 45–64 years, showed that every 100 mg increase in the dietary intake of AA (estimated using a FFQ) was associated with an increase femoral BMD (total and neck, assessed by Dual Energy X-ray Absorptiometry, DXA) equal to 0.017 g/cm^2^. Then stratifying the sample based on dietary calcium intake (> or ≤ 500 mg/day), the association between femoral and lumbar BMD and AA intake was significant only in women with a calcium intake of at least 500 mg/day [25]. More recently Ilich et al., on a sample of 136 healthy postmenopausal women, with an average age of 68.7 ± 7.1 years, showed a significant relationship between the overall intake of AA (through diet, evaluated through a three-day food diary, and through the use of supplements) and femoral BMD (total femur, trochanter, Ward’s triangle and shaft, evaluated by DXA), in multiple regression models controlled for age, fat and lean mass, physical activity and energy intake [31]. On the other hand, taking into account the elderly participants of the Framingham Osteoporosis Study (334 men and 540 women, mean age 75 years, undergoing BMD evaluation by DXA and a 126-item FFQ), a negative association was described between the dietary intake of AA, total and via supplements, and trochanteric BMD in smoking male subjects, and a positive association between total AA intake and femoral neck BMD in non-smoking males. A high overall intake of this vitamin was also correlated with a lower loss of femoral bone density, during an observation period of 4 years, in men with concomitant low intakes of calcium (<661 mg/day) and vitamin E (<7.7 mg/day); the attenuation of the significance of these associations following adjustment of the data for potassium intake (indicator of fruit and vegetable intake) suggests, however, that the effects of AA may not be easily distinguished from those of other factors’ protective substances contained in foods of plant origin. However, no association emerged in female participants [29]. Furthermore, in a cross-sectional case-control study of 144 postmenopausal Korean women aged 50–70, it was found that dietary AA intake (estimated by FFQ) was associated with a significantly reduced risk of osteoporosis (Lumbar, femoral neck or total femoral neck T-score < −2.5 on evaluation by DXA) [37]. Another Korean study, from 2016, also investigated the association between dietary intake of AA and the presence of osteoporosis (diagnosed by DXA) in 3047 adults aged 50 and over: the prevalence of osteoporosis decreased with increasing vitamin intake levels, with participants in the lower quartile of dietary intake being more likely to develop this condition than the upper quartiles [38]. In a retrospective cross-sectional study of 1892 women, aged 55 to 80, who had previously undergone DXA bone densitometry and for whom information on risk factors for osteoporosis, as well as data regarding the intake of AA through diet and supplements, there was no significant correlation between BMD and intake of this vitamin. Only the use of supplements for at least 10 years (mean total intake of 407 mg/day) in younger women (55–64 years) and not on estrogen replacement therapy was associated with higher BMD values [27]. In addition, concomitant intake of estrogen and AA was associated with significantly higher BMD levels at all measurement sites (ultradistal radius and shaft, hip and lumbar spine), and women with concomitant calcium, estrogen and AA intake showed the highest BMD values at the femoral neck, total hip, ultradistal radius, and lumbar spine [39]. Longitudinal studies have also explored the possible relationship between AA and bone health in elderly subjects. In addition to the aforementioned study by Sahni S. et al. of 2008 [29], that of Kaptoge and colleagues on 470 men and 474 women, aged between 67 and 79 years, living in the community, estimated the annual percentage loss of total femoral BMD (assessed by DXA) over an average period of three years (range 2–5 years), showing, in female participants only, how a low intake of AA (estimated through a seven-day food diary) was associated with a more rapid loss of bone density. Specifically, women in the lower tertile (dietary AA intake of 7–57 mg/day) showed an average rate of BMD loss of −0.65% per year, significantly faster (about twice), compared to that of participants in the intermediate tertile (58–98 mg/day, −0.31% per year) and higher (99–363 mg/day, −0.30% per year) [26]. The study by Sahni et al. in 2009, however, performed on participants in the Framingham Osteoporosis Study (aged between 67 and 95 years), evaluated the association between the intake (total, dietary and supplementation, by means of FFQ) of AA and the occurrence of fractures hip and non-vertebral osteoporotic fractures, in a total of 929 and 918 subjects, during a long follow-up period of 17 and 15 years, respectively. Subjects in the upper tertile of total AA intake showed a significantly lower risk of non-vertebral hip and osteoporotic fractures than subjects in the lower tertile; a similar correlation was also found for participants with the highest intake of AA via supplements compared to those who did not use them, even if only the data on hip fractures reached statistical significance. With regard to the intake of AA through the diet alone, a protective trend was observed, although not statistically significant, against both types of fracture [30]. Only one study has not shown a correlation between dietary intake of AA and bone health. In a sample of 11,068 women between the ages of 50 and 79, enrolled in the context of the multicentre Women’s Health Initiative Observational Study and Clinical Trial, no association was observed between AA intake (with diet, assessed by FFQ semi -quantitative, and through supplementation, assessed by an interviewer via questionnaire) and femoral, vertebral and total body BMD (measured by DXA). However, there was a significant interaction between the effects of total AA intake and the use of hormone therapy, as the benefits of such therapy on BMD were greatest in women with the highest vitamin intake [34]. Considering, instead, a sample of healthy pre-menopausal women, therefore younger (994 subjects aged between 45 and 49 years), New et al. found a significant difference in BMD values at the lumbar spine between the lower quartile (lowest BMD) and the upper quartile of dietary AA intake (evaluated, also in this case, by FFQ) [32]. The same authors, in a subsequent cross-sectional study on 62 healthy women aged 45–55 years of age, found a negative relationship between the dietary intake of AA and bone resorption, calculated by measuring the urinary excretion of pyridinoline. and deoxypyridinoline. In particular, the mean excretion of deoxypyridinoline was significantly lower in women with higher AA intake [33]. Starting from the search for a possible association between bone mineral status and intake of fruit and vegetables in a large population, including 212 adolescents (16–18 years of age), 90 young women (23–37 years) and 134 elderly subjects (60–83 years), cross-sectional study by Prynne and colleagues showed a significant positive relationship between the dietary intake of AA (estimated through a 7-day food diary) and BMD and BMC (bone mineral content), assessed by DXA, in all the skeletal sites investigated (lumbar spine, left femur and total body), but only in adolescent males [35]. Based on data collected through the NHANES III between 1988 and 1994, on 13,080 adults aged between 20 and 90 years (3778 pre-menopausal women, 3165 postmenopausal women and 6137 men), Simon and Hudes identified an independent positive association between dietary intake of AA (calculated by 24-h recall, but without investigating the possible use of supplements) and proximal femur BMD (measured by DXA) in premenopausal women, and a “non-linear association between dietary intake of AA and self-reported fractures in men (with the lowest prevalence of fractures for an intake of about 200 mg/day, and higher prevalence values for intakes both below 100 and above 250 mg/day); in postmenopausal women, however, no correlation was found between AA and BMD intake or self-reported fractures [36].

### 4.3. Effects of AA on Bone Metabolism in Humans: Intervention Studies

Some clinical trials have been developed in order to investigate the relationship between the intake of AA supplements (alone or in combination with other nutrients) and bone health.

In order to investigate the possible relationship between the daily use of AA supplements and BMD, 994 postmenopausal women were recruited (50–98 years of age, sample average of 72 years), 277 of whom regularly take these supplements, with a daily intake of between 100 and 5000 mg (average dose of AA equal to 745 mg) for an average period of 12.4 years. After adjusting the data for age, BMI and total calcium intake, BMD values in the radial shaft, femoral neck and total hip were found to be approximately 3% higher in women who took supplements [39]. In a study by Pasco and colleagues, the duration of use of antioxidant supplements (AA and/or vitamin E), found in 22 of the 533 postmenopausal non-smoking women recruited, was negatively associated with serum levels of telopeptide C (CTx), a biochemical marker of bone turnover, suggesting how supplementation with antioxidant vitamins (C and E) can repress bone resorption [28]. In a 2007 clinical study, 75 osteoporotic patients, aged between 45 and 70 years, were divided into three intervention groups of 25 subjects each: group A, recipient 400 mg of vitamin E/day, group B, recipient 500 mg of AA/day, and group C, supplemented with both vitamins. After a 90-day supplementation period, it was found that supplementation with AA reduced serum levels of tartrate-resistant acid phosphatase (TRAP, marker of osteoclastic activity) and improved the body’s antioxidant status (increased superoxide dismutase, SOD, and erythrocyte glutathione—GSH—levels), similar to what was also observed in the group treated with vitamin E alone the combination of the two vitamins, on the other hand, while determining a further improvement in the oxidative state, did not seem to influence the concentrations of TRAP. These observations would suggest an overall improvement in bone status due to prolonged integration with antioxidant vitamins, administered individually or in combination [42]. In a randomized controlled pilot study, 34 healthy postmenopausal women (mean age of 66.1 ± 3.3 years) were divided into four groups: placebo (*n* = 7), antioxidant administration (600 mg of vitamin E + 1000 mg of AA per day, *n* = 8), placebo + endurance exercise (three times/week, *n* = 11) and antioxidants + exercise (*n* = 8). After a period of six months, the authors observed a significant reduction in lumbar BMD (assessed by DXA), but not in femoral neck, in the placebo group compared to baseline, while in the other three groups the values were stable in both locations. While suggesting such supplements as antioxidant vitamins (E and C) may exert some protective effect against bone loss, similar to endurance exercise, the combination of these two elements did not show additional effects, thus making necessary further investigations to understand their possible role in the prevention of osteoporosis [40]. In another randomized, controlled, double-blind clinical trial, 90 elderly subjects were assigned to three different intervention groups (30 participants for each group): Tx0 (placebo), Tx1 (500 mg AA + 400 IU of α -tocopherol (vitamin E) per day), and Tx2 (1000 mg of AA + 400 IU of α-tocopherol per day). After 12 months, less hip bone loss (BMD assessed by DXA) was found in the Tx2 group compared to Tx0, suggesting that daily administration of 1000 mg of AA and 400 IU of vitamin E could prove useful in prevention and treatment of age-related osteoporosis [41]. Maïmoun et al. also investigated the effects of daily supplementation with 500 mg of AA and 100 mg of vitamin E, in association with programmed aerobic exercise, on bone metabolism in 13 healthy elderly individuals (mean age equal at 74 years old). After 8 weeks, serum concentrations of 25-hydroxyvitamin D and 1,25-dihydroxyvitamin D were significantly increased, while parathyroid hormone levels had decreased; according to the authors, such modifications could serve to counterbalance the unfavorable hormonal profile often observed in the elderly and responsible for accentuated bone loss [43]. In contrast to the results described above appear to be those of the randomized, placebo-controlled, double-blind study developed by Stunes and colleagues, who evaluated the skeletal effects of high-dose supplementation with AA (1000 mg/day) and vitamin E (235 mg/day) in 33 healthy men over 60 years of age during 12 weeks of controlled resistance training. In the control group (*n* = 17; exercise only) there was a 1% increase in total hip BMD (measured by DXA) from baseline, and a significant 0.9% increase in lumbar BMD compared to the treated group, also with AA and vitamin E (n = 16; 1.0% of BMD). The authors therefore hypothesized potential adverse effects of high doses of antioxidant vitamins on bone health, as these nutrients could limit the positive effects obtained through resistance exercise, and recommend caution in their use, especially in non-healthy individuals. [44]. However, there are no other studies in the literature that confirm these results.

## 5. Conclusions

The aim of this narrative review was to consider state of the art on correlation between BMD and AA dietary intake and AA blood levels and on the effectiveness of AA supplement in humans. Specifically, 15 studies have been published in the literature on the correlation between AA intake and bone mineral density: eight studies that demonstrated the correlation between AA dietary intake and BMD in menopausal women (in total 9664 menopausal women). In only one study was there no association observed between total AA intake and BMD by DXA. However, there was a significant interaction between the effects of total AA intake and the use of hormone therapy, given that the benefits of this therapy on BMD were greater in women with the highest vitamin intake. Even in premenopausal women the data was confirmed: the two studies that evaluated the correlation (1056 women), demonstrated significance. Finally, the two studies (13,516 subjects) that considered the correlation between AA intake and BMD in different age groups starting from adolescence identified an independent positive association between dietary intake of AA (calculated by 24-h recall, but without investigating the possible use of supplements) and BMD of the proximal femur (measured by DXA) in premenopausal women, and a non-linear association between dietary intake of AA and self-reported fractures in men (with the lowest prevalence of fractures for an intake of approximately 200 mg/day, and higher prevalence values for intakes of both less than 100 and greater than 250 mg/day)

Considering the studies that evaluated the blood concentration of AA and the BMD, there are four, with a total of 337 patients, and all confirm the positive correlation. One study also found a linear association between serum AA and fracture risk, significantly reduced in subjects with serum values in the upper quartile.

Regarding the studies that evaluated supplementation, there are 6 (2671 subjects), of which only one was carried out with supplementation exclusively of AA, while in the others the supplementation considered a set of antioxidants (AA and vitamin E)

In the study that evaluated supplementation with AA alone, 994 postmenopausal women (50–98 years of age, sample mean 72 years) were recruited, of which 277 regularly took these supplements, with a daily intake between 100 and 5000 mg (average dose of AA equal to 745 mg) for an average period of 12.4 years. After adjusting the data for age, BMI and total calcium intake, BMD values in the radial shaft, femoral neck and total hip were found to be approximately 3% higher in women who took supplements. The other four studies in which patients took AA (range 500–1000 mg/day) together with vitamin E confirmed significant positive activity on bone health demonstrated by evaluating BMD values in three studies (in total 1118 patients), in one study the increase of vitamin D (13 patients), and in one study the activity of telopeptide (546 patients).

Given the results of this narrative review, it is worth paying attention to the advantageous impact of a proper AA intake in the diet on bone metabolism because it is plausible to date the existence of a positive association between dietary intake of AA and bone mineral density. This association however, appears complex, as it can be influenced by the interaction with other factors, such as cigarette smoking, menopausal estrogen hormone therapy and dietary calcium intake. Considering studies on AA blood concentration and BMD, there are four (337 patients) that confirm positive correlation. Regarding studies on supplementation, there were 6 (2671 subjects), of which one was carried out with AA supplementation exclusively in 994 postmenopausal women with a daily average dose of 745 mg (average period: 12.4 years). BMD values were found to be approximately 3% higher in women who took supplements.

## Figures and Tables

**Figure 1 nutrients-13-01012-f001:**
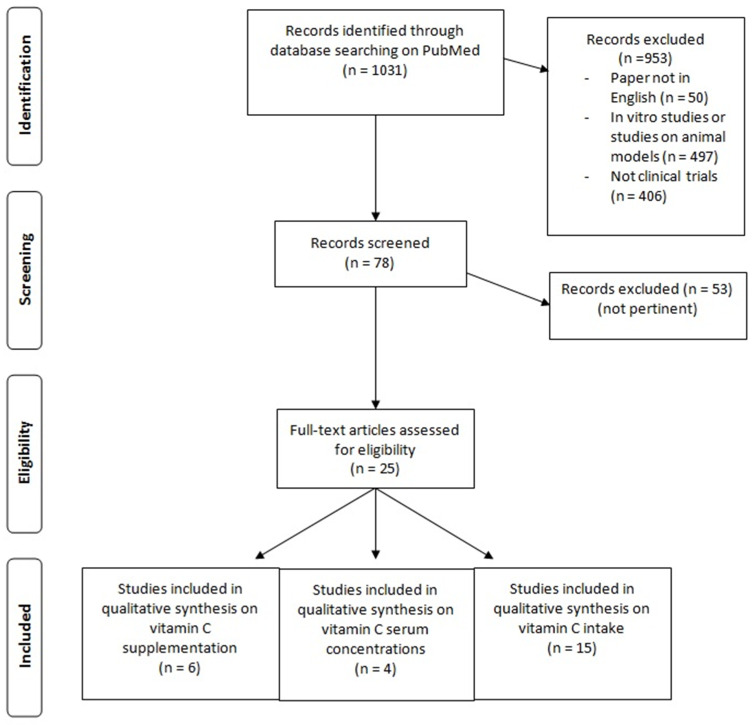
Flow chart of the literature research.

**Table 1 nutrients-13-01012-t001:** Pathophysiological key messages on the role of AA in bone metabolism.

Authors, Year [Reference]	Type of Study	Results
Aghajanian P., 2015 [7]	Animal model (rats and mice)	Vertebrate organisms deficient in AA develop bone disorders
Kipp D.E., 1996 [8]	Animal model (guinea pigs)	Scurvy but not food restriction, per se, results in alterations in bone mass and tissue collagen synthesis
Tsunenari T., 1991 [9]	Animal model (AA deficent-rats)	Osteopathy could be due to AA deficiency itself rather than malnutrition
Bates C.J., 1979 [10]	Animal model (guinea pigs)	Differential concentration of AA by different tissues seems more likely to be the critical factor
Tsuchiya, H., 1994 [11]	Animal model (guinea pigs)	Low tissue AA levels in guinea pigs alter the connective tissue composition of bones
Leboy P.S., 1989 [12]	In vitro	AA play a role in endochondral bone
Daniel, J.C., 1984 [13]	In vitro	AA facilitates the formation of an extracellular matrix in chondrocyte cultures
Temu, T.M., 2010 [14]	In vitro	AA promote the differentiation of ATDC5 cells
Franceschi, R.T. 1992 [15]	In vitro	Actions of AA on osteoblast marker gene expression are mediated by increases in collagen synthesis
Franceschi, R.T. 1994 [16]	In vitro	AA addition that allows subsequent induction of osteoblast-related genes
Tsuneto, M., 2005 [17]	In vitro	AA promote osteoclastogenesis of ES cells
Xiao, X.H., 2005 [18]	In vitro	AA inhibits receptor-activated nuclear factor kappaB ligand (RANKL)-induced differentiation of osteoclast
Le Nihouannen, D., 2010 [19]	In vitro	During osteoclastogenesis AA acts as an oxidant, first stimulating osteoclast formation, but later limiting osteoclast lifespan

**Table 2 nutrients-13-01012-t002:** Studies that considered blood ascorbid acid (AA) levels.

First Author, Year [Reference]	Study Design	Setting	Number of Subjects (M-F) Mean Age	Primary Outcomes	Micronutrient Serum Concentration Osteoporosis	Micronutrient Serum Concentration Osteopenia	Micronutrient Serum Concentration Normal	Micronutrient Serum Reference Value	% Subjects < Reference Value	Results
Maggio D., 2003 [21]	Cross-sectional, case-control study	Free-living subjects	150 F 75 osteoporosis (70.4 ± 8.5 y) 75 controls (68.8 ± 3.5 y)	Plasma AA, vitamin E, and A; uric acid; superoxide dismutase, glutathione peroxidase	Plasma AA: 30 ± 3.7 μmoL/L	//	Plasma AA: 55.5 ± 13.1 μmoL/L	//	None of the subjects belonging to the two groups had levels below the normal AA range.	Dietary and endogenous antioxidants consistently lower in osteoporotic than in control subjects
Falch J.A., 1998 [22]	Case-control study	Cases admitted to the hospital and home-living controls	40 81 ± 5 y (20 cases with a recent hip fracture + 20 controls)	Serum AA concentration	Patients with a recent hip fracture: 34 ± 19 μmoL/L	//	Controls: 54 ± 30 μmoL/L	//	//	Serum AA concentration significantly lower in the hip fracture patients.
Martínez-Ramírez M.J., 2007 [23]	Hospital-based case-control study	Hospital of Jaén, Spain	334 (167 cases + 167 controls) (80% F)	Osteoporotic fractures	In cases: 19.31 μmoL/L	//	In controls: 23.28 μmoL/L	//	//	Statistically significant difference between cases and controls for AA blood levels. Association between serum AA and fracture risk, with a significantly reduced risk for the upper quartile.
Lumbers M., 2001 [24]	Case-control study	Cases: hospital patients admitted for emergency surgery for fractured neck of femur. Controls: independent-living females	125 F 75 cases (80.5 y) 50 controls (79.8 y)	Levels of plasma albumin, transferrin, C-reactive protein (CRP), cholesterol, AA, Se, Zn, total antioxidant status, Se-dependent glutathione peroxidase activity.	Hip fracture patients Plasma AA: 42.7 ± 21.4 μmoL/L	//	Controls Plasma AA: 20.8 ± 14.2 μmoL/L	//	//	Fracture patients: higher plasma AA levels.

**Table 3 nutrients-13-01012-t003:** Dietary intake of AA and bone metabolism.

First Author, Year [Reference]	Study Design	Setting	Number of Subjects (M-F) Mean Age	Lowest Quintile Intake/RDA or EAR	% Subject in Lowest Quintile Intake/% Subject < RDA or EAR	Highest Quintile Intake	% Subject in Highest Quintile Intake	Primary Outcomes	Results
Hall SL, 1998 [25]	randomized, double-blinded, placebo-controlled study	Data from PEPI study	775 F aged 45–64 y	//	//	//	//	Cross-sectional relation between dietary AA intake and BMD.	Positive association between AA and BMD. 0.018 g/cm^2^ BMD increment for each AA additional 100 mg intake
Kaptoge S, 2002 [26]	Longitudinal study	Data from EPIC Norfolk study	470 M–474 F aged 67–79 y	lowest tertile (7–57 mg/day)	//	upper tertile (99–363 mg/day)	//	Nutritional determinants of BMD loss from the hip in a community-based sample	Women in the lowest AA tertile intake lost BMD at an average rate of −0.65%, significantly faster compared to loss rates in the middle (−0.31%) and upper intake tertiles (−0.30%)
Leveille SG, 1997 [27]	Cross sectional study	Seattle area of Washington State	1892 F 55–80 y	//	//	//	//	Relationship between dietary AA and hip BMD in postmenopausal women	No BMD differences according to diet-only AA intake or combined dietary AA and supplementation. Women who used AA supplements for >10 y had a higher BMD than non-users
Martínez-Ramírez MJ, 2007 [23]	Case-control study	Hospital of Jaén	167 cases–167 controls	//	//	//	//	Influence of water-soluble vitamins on adequate bone tissue structure development	The AA intake has not been related to fracture risk
Pasco JA, 2006 [28]	Observational study	Barwon Statistical Division surrounding Geelong in southeastern Australia	533 F	//	//	//	//	Associations among use of antioxidants, AA and vitamin E, serum bone turnover markers and BMD	Antioxidants, vitamin E or AA supplements may suppress bone resorption
Sahni S, 2008 [29]	Population-based cohort study	Data from Framingham Osteoporosis Study	334 M–540 F 70–80 y	Lowest tertile: 80–160 mg/day	//	Highest tertile: 242–314 mg/day	//	Associations of supplemental/dietary AA intake with BMD at varius sites and 4-y BMD change	Higher dietary AA intake tended to be associated with lower femoral neck-BMD loss
Sahni S, 2009 [30]	Population-based cohort study	Data from Framingham Osteoporosis Study	366 M–592 F 70–80 y	Lowest tertile (median: 94 mg/day)	//	Highest tertile (median: 313 mg/day)	//	Possible protective effect of AA on bone health	Subjects in the highest tertile of total AA intake had significantly fewer hip and non-vertebral fractures
Ilich JZ, 2003 [31]	Cross-sectional study	Eastern part of Connecticut	136 F 68.7 ± 7.1 y	//	//	//	//	Relationship between various nutrients and BMD of several skeletal sites	AA has been significantly related to BMD of several skeletal sites.
New SA, 1997 [32]	Cross-sectional study	Data from Osteoporosis Screening Program—Aberdeen (Scotland)	994 F, 45–49 y	40 mg/day (EAR)	//	//	//	Association between dietary intake and BMD	The BMD across the quartiles of AA intake was nonlinear. BMD was higher in the third quartile and significantly different from the lowest quartile at all four sites even after adjustment for the confounding factors
New SA, 2000 [33]	Cross-sectional study	Data from Osteoporosis Screening Program—Aberdeen (Scotland)	62 F, 45–55 y	40 mg/day (EAR)/	//	//	//	Association between micronutrients identified as important to BMD and bone heath indexes	A nonsignificant trend was seen for AA intake and deoxypyridinoline excretion. Low intakes of AA were associated with increased bone resorption
Wolf RL, 2005 [34]	Population-based cohort study	3 clinic sites (Pittsburgh, PA; Birmingham, AL; Tucson, AZ) of Women’s Health Initiative (WHI)	11,068 F, 50–79 y	//	//	//	//	Association between higher/total intakes, and serum antioxidants with higher BMD	A significant interaction effect has been demonstrated between intake of total AA and use of hormone therapy
Prynne CJ, 2006 [35]	Cross-sectional study	Cambridge (United Kingdom)	436 (M-F) 16–83 y	Lowest tertile: 46–80 mg (boys) and 46–68 mg (girls)		Highest tertile: 134–181 mg (boys) and 124–169 mg (girls)		Associations between BMD and actual fruit and vegetable intakes, as estimated from 7-d food diaries	In boys only, femoral neck size-adjusted bone mineral content was significantly and positively associated with the intakes of both fruit and dietary AA
Simon J, 2001 [36]	Population-based cohort study	Data from NHANES III during 1988–1994	13080 (3778 premenopausal women, 3165 postmenopausal women, and 6137 men) 26–75 y	//	//	//	//	Relation of AA to BMD and the prevalence of self-reported fractures	Dietary AA intake was independently associated with BMD among premenopausal women. Among men dietary AA intake was associated in a nonlinear mode with self-reported fracture tucker
Park HM, 2011 [37]	Case-control study	10 different hospitals in Seoul	144 F (72 cases, 59.76 ± 0.5 y—72 controls 58.03 ± 0.66 y)	≤ 91.54 mg/day (lowest quartile)	32 cases—18 controls	9 cases–18 controls	> 176.30 mg/day (highest quartile)	Examine the hypothesis that calcium from vegetable sources is associated with osteoporosis risk and BMD	Intake of vegetables and some nutrients as AA was associated with significantly reduced risk of osteoporosis
Kim MH, 2016 [38]	Case-control study	Data from the 2008, 2009, 2010, and 2011 Korean National Health and Nutritional Examination Survey (KNHANES)	3047 (M-F) 64.6 ± 0.3 y	0.0–45.0 mg/day (case group)	452 subjects (case group)	128.1–801.5 mg/day (case group)	202 subjects (case group)	Associations between AA intake, physical activity, and osteoporosis	Higher AA intake levels have been associated with a lower risk of osteoporosis in Korean adults aged over 50 with low levels of physical activity

**Table 4 nutrients-13-01012-t004:** Studies that considered AA supplementation.

First Author, Year [Reference]	Study Design	Setting	Intervention	Parallel Treatments	Number of Subjects (M-F)	Duration of the Intervention	Primary Outcomes	Secondary Outcomes	Results
Morton D.J., 2001 [39]	Population-cohort study	Free-living subjects	Use of daily AA supplements: from 70 to 5000 mg/day (mean 745 mg/day)	Non use of AA supplements	994 F 50–98 y (mean age 72 y)	Mean duration 12.4 y	BMD at the ultradistal radius and midshaft radius of the nondominant arm using single photon absorptiometry; femoral neck, total hip, and lumbar spine BMD by DXA.	//	In AA users BMD levels approximately 3% higher at the midshaft radius, femoral neck, and total hip.
Chuin A., 2009 [40]	Pilot randomized, controlled study	Free-living subjects	Antioxidants (n = 8) (600 mg/day vit E + 1000 mg/day AA)	Placebo (n = 7); resistance exercise (supervised 60-min sessions 3 times/week on alternating days) and placebo (n = 11); resistance exercise and antioxidants (n = 8).	34 F 61–73 y (mean age 66.1 ± 3.3 y)	6 months	Femoral neck and lumbar spine BMD (DXA)	//	Significant decrease in the placebo group for lumbar spine BMD, stable in all other groups. No changes for femoral neck BMD.
Ruiz-Ramos M., 2010 [41]	Randomized, double-blind, controlled clinical trial	Free-living subjects	Group Tx1 (n = 30): 500 mg AA + 400 IU vit E/day; group Tx2 (n = 30): 1000 mg AA + 400 IU vit E/day	Placebo (group Tx0, n = 30)	90 (25 M-65 F)	12 months	Thiobarbituric acid reactive substances, total antioxidant status, superoxide dismutase, glutation peroxidase; BMD of hip and spine (DXA).	//	Statistically significant positive correlation between hip-BMD and SOD activity and of GPx. In terms of BMD, less bone loss at the hip level in group Tx2 vs. group Tx0.
Chavan S.N., 2007 [42]	Randomized controlled study	Free-living subjects	500 mg/day AA (group B, n = 25)	400 mg/day vit E (group A, n = 25); 500 mg/day vit C + 400 mg/day vit E (group C, n = 25).	75	90 days	Serum alkaline phosphatase, free or ionic calcium, inorganic phosphorus; tartrate resistant acid phosphatase, malondialdehyde; superoxide dismutase and erythrocyte reduced glutathione.	//	In group A: significant serum MDA and TrACP decrease after 45 and 90 days, non significant serum ALP decrease. In group B: significant decrease of serum MDA and TrACP after 45 days and 90 days, significant decrease of serum ALP only at 90 days.
Maïmoun L., 2008 [43]	Observational study	Free-living subjects	AA (500 mg) and vit E (100 mg) daily + supervised progressive aerobic training programme	//	13 (mean age 73.9 ± 3.8 y) (4 M-9 F)	8 weeks	Calcium homeostasis, bone cell activity (peripheral bone biochemical markers), bone related hormones.	//	25-hydroxyvitamin D and 1,25-dihydroxyvitamin D concentrations significantly increased by 42.8% and 26.8% respectively; parathyroid hormone concentration decreased by 17.5%. Bone alkaline phosphatase decreased by 14.6%.
Stunes A.K., 2017 [44]	Double-blinded, randomized, placebo-controlled study	Free-living subjects	1000 mg AA+ 235 mg vit E daily (antioxidant group, AO, n = 16) + supervised strength training	Placebo (control group, CG, n = 17) + supervised strength training	33 M	12 weeks	Areal BMD at whole body, lumbar spine, total hip, and femoral neck (DXA), muscle strength by 1 RM.	Serum analyses of bone-related factors and adipokines.	In the controls: total hip aBMD increased by 1.0% versus pretest, and lumbar spine aBMD increased by 0.9% compared to the supplemented group.

## Data Availability

The data presented in this study are available in the text.

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
