# Peer review of "Evidence of a Positive Link between Consumption and Supplementation of Ascorbic Acid and Bone Mineral Density"

_nutrients, 2021, doi:10.3390/nu13031012_

Round 1

Reviewer 1 Report

Overall this is a well-explained study describing a review of a reasonably discrete number of papers that have examined the potential for a role of vitamin C (AA) in bone health. The descriptions of the studies are clear. The conclusions of this paper are not particularly novel in terms of clinical studies of AA supplementation having only mild effects and in only some groups, but they fit well with the literature of other roles of AA as observed in clinical populations.  There is likely a strong role for methodology of the original studies in this result as the links between blood, intake and supplementation are not alays as clear as could be wished in population based studies..

It would probably help clarify the impact of this work if the authors would include a figure or brief schematic to identify how and where AA may exert its actions in bone health. Several different mechanisms are described briefly in discussion and introduction but none is fully fleshed out. A figure would help show how it all fits together.

Levels of AA in blood, where known, were not particularly high, and in several studies were all within normal ranges, even if there are differences between the groups. It is likely that these levels can be achieved through diet and supplementation is not necessary – which is different from the case where there is a need to correct a deficiency. It is important to clarify this and perhaps point out more strongly any weaknesses in the original studies in this regard.

Minor/editing points

Authors may need to check their references in the introduction to improve accuracy. For example the first reference is the 1933 Haworth paper which likely does not include “the prevention of certain cancers, cardiovascular diseases and disorders involving oxidative stress“.

The layout of table 1 is less than ideal because of the many boxes with so much text. Some entries stretch across 2 pages by a few words. Better formatting is needed.

Most but not all levels are reported as umol/L, would be very helpful to include conversion for the few studies that are not

P22-24 is one giant paragraph and should be broken up

Reviewer 2 Report

Dear Editor,

the narrative review by Rondanelli et al discusses the association between ascorbic acid (nutritional intake, supplements, blood levels) and bone mineral density in humans. The study is well conducted. However, few minor issues may be considered prior to publication.

Title. The term “bone health” could be changed into “bone mineral density”, which looks the primary outcome investigated.

Introduction. Report of experimental data is a bit more extended than what expected in usual introductions. This session could be synthesized and replaced within discussion. Dedicated figure may help readers in catching pathophysiological key messages just as quicky.

Table 1-3 are very hard to be followed, mainly due to extension, amount of information and editing. Authors may consider to split variables into more tables or to revisit editing.

A fourth table could be added, to summarize a) Authors’ conclusions about evidence based association, existing between BMD and nutritional intake, supplements and circulating levels of ascorbic acid, b) clinical implications of results, c) future perspectives on the topic under Authors’ opinion.

Reviewer 3 Report

The review article is well prepared and has a relevant topic, however a paragraph must be added indicating the limitations of the review study, as well as the limitations of the studies used to make the review article. It should also be added to the conclusion what are the needs for studies to be done in the future on this topic.
